# Job Demand-Control-Support Model as Related to Objectively Measured Physical Activity and Sedentary Time in Working Women and Men

**DOI:** 10.3390/ijerph16183370

**Published:** 2019-09-12

**Authors:** Kristina Larsson, Örjan Ekblom, Lena V. Kallings, Maria Ekblom, Victoria Blom

**Affiliations:** 1The Swedish School of Sport and Health Sciences, 11486 Stockholm, Sweden; orjan.ekblom@gih.se (Ö.E.); lena.kallings@gih.se (L.V.K.); maria.ekblom@gih.se (M.E.); victoria.blom@gih.se (V.B.); 2Sophiahemmet University, 11486 Stockholm, Sweden; 3The Department of Neuroscience, Karolinska Institutet, 17177 Stockholm, Sweden

**Keywords:** job demand-control-support model, physical activity, sedentary time, workplace, activPAL, ActiGraph, questionnaire

## Abstract

A physically active lifestyle incurs health benefits and physically active individuals show reduced reactivity to psychosocial stressors. However, the findings are inconclusive and are based on self-reported physical activity and sedentary time. The present study aimed at studying the associations between psychological stressors (job demand, control, support, JD-C-S) and objectively measured physical activity (PA) on various intensities from sedentary (SED) to vigorous physical activity. The participants were 314 employees from a cross-sectional study. PA data were collected with the accelerometer ActiGraph GT3X (Pensacola, FL, USA), SED data with the inclinometer activPAL (PAL Technologies Ltd., Glasgow, Scotland, UK), and psychosocial stressors with a web questionnaire. Results showed that vigorous-intensity PA was negatively associated with demand (β −0.15, *p* < 0.05), even when adjusted for the covariates. SED was negatively associated to support (β −0.13, *p* < 0.05). Stress significantly moderated relations between support and sedentary time (β −0.12, *p* < 0.05). Moderate PA (MVPA) was negatively associated with demand, but only when controlling for overtime (β −0.13, *p* < 0.05). MVPA was also negatively associated with control (β −0.15, *p* < 0.05) but not when work engagement was included in the model. Being more physically active and spending less time sedentary may help to handle job situations with high demand and low support.

## 1. Introduction

There is an increasing interest in improving workplace health by promoting physical activity in order to increase sustainability and reduce stress reactions when exposed to work stressors. There is substantial evidence in support of a preventive effect of physical activity on ill-health [1]. Some studies show that individuals who perform physical activity before exposure to stress perceive less intense stress and there is also some support for physically active individuals improving their recovery after stress exposure [2,3]. Stress reduction has also been recognized as a benefit of regular moderate-intensity exercise [4]. To our knowledge, sitting time has not been found to be associated with stress [5] but lower levels of physical health were observed in a group with more time spent sitting [6]. However, previous studies often used subjective methods of sitting time and physical activity, known to suffer from considerable limitations concerning the recall and reporting bias, leading to limited validity and precision [7,8].

Stressors in the psychosocial work environment are often measured with the job demand-control-support model, JD-C-S [9,10], see Figure 1. Perceiving having a job situation with high demand in combination with low control to meet up to these demands (i.e., job strain situation) has been shown to be related to cardiovascular disease, depression, and stress-related ill-health [11,12]. In contrast, a situation with high control and high job demand (i.e., active job situation) has been shown to be associated with work engagement and good health [13,14]. There is also some evidence for a moderating role of support in reducing the effect of high demand and low control [13,14]. Managers have shown to perceive higher demand and control in their work than non-managers, and female managers have shown to be particularly at risk for burnout, facing higher demands which are not counterbalanced by a higher control as commonly seen for their male counterparts [15].

A few studies have investigated associations between physical activity and the job-demand-control support model. In cross-sectional analyses Fransson and colleagues found that employees with high-strain and passive jobs had 21–26% higher odds for physical inactivity during leisure time compared to employees in low-strain jobs. In the same study, prospective analyses showed that the odds of becoming physically inactive during follow-up after 2–9 years were 21% and 20% higher for those with high-strain and passive jobs at baseline [16]. A Finish population study among men and women in the public sector also found that high job strain was associated with lower leisure-time physical activity [17]. Hansen and colleagues found that vigorous activity moderated the association between perceived job demand-job control and perceived stress and energy. However, the same study showed that physically active employees did not report less perceived job demand and higher perceived job control compared to non-physically active employees [18].

Stress and time have been reported as barriers to being physically active [19]. Physical activity may be reduced when stressed due to symptoms related to stress, such as fatigue or low mood or due to high workload in order to save time [20]. Worth taking into account is also the concept of work engagement, often defined as a positive, fulfilling, work-related state of mind that is characterized by vigor, dedication, and absorption [21] which may be related to both physical activity, job stressors, and resources. Physical activities have been shown to increase next morning work engagement vigor through enhanced psychological detachment and relaxation [22] and job resources such as job control may increase work engagement [21]. However, work engagement may also absorb the individual in there work and, thereby, cause long working hours and thus reducing the time for physical activity.

The findings on work stressors in terms of JD-C-S model and physical activity are based on a few studies using self-reported physical activity (PA). The present study aimed to assess the association between each of the components of the JD-C-S model and objectively measured PA on various intensities from sedentary time (SED) to vigorous PA and the extent to which these associations were affected by variables such as manager role, overtime work, stress, work engagement, age, gender, and education.

Hypotheses: (1) High demand will be negatively associated with PA and positively associated with SED, (2) high control will be positively associated with PA and negatively associated with SED, (3) high support will be positively associated with PA and negatively associated with SED.

## 2. Materials and Methods

### 2.1. Participants and Sampling

The data in this study were retrieved from a larger project with a cross-sectional design called Physical Activity and Healthy Brain Functions during 2016–2017. The project was performed at The Swedish School of Sport and Health Sciences (GIH). The Stockholm regional ethical review board approved the project (Dnr 2016/1840-32) and all participants signed a written informed consent form.

Participants were office-based employees at three workplaces in Stockholm and Gothenburg who were invited to participate via mail (*n* = 2024) to respond to a self-reported web-based questionnaire. Two weeks later, they attended a test session in which they were equipped with two activity monitors (activPAL (PAL Technologies Ltd., Glasgow, Scotland, UK) and ActiGraph GT3X (Pensacola, FL, USA)) and received a diary log. The participants wore the activity monitors and noted the time points for when they went to bed and woke up, for seven consecutive days.

### 2.2. Questionnaire

The web-based questionnaire assessed JD-C-S based on sub-questions included in the original model (see Appendix A), which was calculated to index based on the mean value for the subscales demand, control and support [9]. The internal consistency of the subscales has previously shown generally satisfactory alfa-levels (demand α = 0.73, control α = 0.74 and support α = 0.83) [23]. The internal consistency of the scales within the current sample showed lower alfa-levels for demand (α = 0.32) and control (α = 0.55), and equal level for support (α = 0.88). Items 4 and 9 (see Appendix A) were reversed before the calculation. The questionnaire also included one question to assess overtime work with five answer options (No, Yes 1–3 h/week, Yes 3–5 h/week, Yes 5–10 h/week, Yes > 10 h/week) [24]. Stress was measured with the single item question “Stress means a situation in which a person feels tense, restless, nervous or anxious or is unable to sleep at night because his/her mind is troubled all the time. Have you felt this kind of stress for the last few weeks?” The response was recorded on a five-point scale varying from “Not at all” to “Very much” [25]. Work engagement was assessed with the single item question “I get carried away when I am working”, with answer options on a five-point scale from “Not true at all” to “Agree completely” [26]. Manager role and demographic information about, age, gender, and length of education were also included in the questionnaire.

### 2.3. Objectively Measured Physical Activity and Sedentary Time

To assess physical activity (PA), the accelerometer ActiGraph GT3X (ActiGraph, Pensacola, FL, USA) was used. The triaxial ActiGraph GT3X was worn on the right hip with an elastic band. The ActiGraph is not waterproof, therefore the participants were instructed to remove the monitor when showering or swimming. Data were recorded at a sampling frequency of 30 Hz and the Sasaki cut-points were used to estimate PA in different intensities [27]. Data were considered valid if there were more than 600 min per day for at least four days.

The inclinometer activPAL (model activPAL3 micro, PAL Technologies Ltd., Glasgow, Scotland, UK) was used to assess sedentary behavior (SED). The activPAL is a triaxial activity monitor which continuously record the orientation of the thigh at a sampling rate of 20 Hz. To waterproof the activPAL, it was placed in a small condom with transparent film around (Tegaderm Roll, 3M, Maplewood, MN, USA), which also was used to attach the activPAL onto the frontal aspect of the midline of the participant’s right thigh. Periods of sedentary (sitting or lying in a reclining posture), standing and walking were identified by merging the raw data (downloaded with activPAL software version 7.2.32) with the time parameters from the diary log. Sleep time was excluded based on the time points noted in the diary logs (using Excel HSC PAL analysis software V2 19s, developed by Dr. Philippa Dall and Professor Malcolm Granat, School of Health and Life Sciences, Glasgow Caledonian University).

### 2.4. Data Analysis

Statistical analyses were carried out using IBM SPSS Statistics version 25 (Armonk, NY, USA). Linear regression analyses were conducted to test for whether objectively measured moderate to vigorous-intensity physical activity (MVPA), at least vigorous-intensity physical activity (VPA) and SED were associated to the components of the JD-C-S-model (job demand, control, and support). The models were adjusted for age, gender, education, management, stress, overtime work, and work engagement. These confounders were chosen due to their relevance to the study. Analyses with light-intensity PA and the components (high strain, low strain, active and passive jobs) of the JD-C-S-model were also conducted. However, they are not further mentioned since they did not show any significant results. Mediation and moderation analyses were conducted using the SPSS PROCESS Macro [28]. Decisions about what mediation and moderation analyses to conduct were based on the results from the regression analyses. If adding a confounder significantly changed the strength of an association in the regression analyses, this confounder was investigated as both a mediator and a moderator using the SPSS PROCESS Macro. Support and stress were dichotomized into high and low levels based on the median value. A general linear model was used to assess differences in SED in participants with high and low levels of support and stress. MVPA and VPA were not normally distributed and therefore a logarithm function was used. The MVPA, VPA, and SED-data were expressed as a percentage of the wear time.

## 3. Results

### 3.1. Participants Characteristics

In total, 794 participants answered the questionnaire and 320 had valid data from the ActiGraph and activPAL. At last, 314 participants were included in the final dataset (66.9% female), with a mean age of 42.2 years (SD 21.0, min 21, max 66). The mean education length was 14.4 years (SD 2.3, min 9.0, max 22.0). In the sample, 16.6% had a management position. Table 1 presents the distribution of the participants based on the variables included in the regression analyses.

### 3.2. Physical Activity, Sedentary Time, and JD-C-S

Table 2 presents the associations between physical activity, sedentary time and JD-C-S. The linear regression analyses showed that higher levels of demand were related to less physical activity of at least vigorous-intensity even when adjusting for the covariates (age, gender, education, management, stress, overtime, and work engagement) (β −0.15, *p* < 0.05).

Higher levels of support were related to less sedentary time, when adjusting for age, gender, education, management, stress, and overtime (β −0.13, *p* < 0.05). When the model did not include stress or overtime, no association was found. Moderation analysis showed that stress significantly moderated the relationship between support and sedentary time (β −0.12, *p* < 0.05). Analyses with general linear model showed that the relationship between higher levels of support and less sedentary time occurred only in the high-stress group. When the regression model was fully adjusted for all the covariates, this relationship disappeared.

Higher levels of demand were related to less physical activity of moderate intensity, but only when overtime was included in the model (β −0.13, *p* < 0.05). However, no mediation or moderation effect was found for overtime.

Higher levels of control were related to less physical activity of moderate intensity, but not when work engagement was included in the model (β −0.15, *p* < 0.05). No mediation effect was found for work engagement.

## 4. Discussion

The aim of the present study was to assess associations between JD-C-S and objectively measured PA and SED, and the importance of manager role, overtime work, stress, work engagement, age, gender, and education for these associations.

In line with our first hypothesis, the results indicate a negative association between job demand and PA at vigorous intensities, where more PA was related to less perceived demand. Jobs requiring high demand (e.g., working hard and fast) might be both time and energy consuming which may relate to less physical activity on higher intensities as seen in planned exercise. Hansen et al. also found an effect of vigorous activity and concluded that vigorous physical activity had a moderating effect on the association between perceived job demand-job control and perceived stress and energy [18]. However, they measured physical activity using self-report.

Higher levels of control were related to less PA of moderate intensity, but not when work engagement was included in the model. This was not expected and contradicted our second hypothesis. However, this may be explained by high control being associated with high levels of responsibility which may reduce the time for PA. This reasoning is in line with the results that higher job demand was related to less vigorous PA. Work engagement may also absorb the individual in her work and thereby cause long working hours, reducing the time available for physical activity. However, mediation analyses showed that work engagement did not significantly mediate the association between control and MVPA.

Our results show that higher levels of perceived support were related to less sedentary time when stress was included in the model. The results are in line with the third hypothesis. A workplace with high social support, a good relationship with colleagues and the manager, may lead to a freer and more flexible atmosphere. This might lead to more breaks for socializing with colleagues during the workday and thereby affect the sedentary time. A situation with low social support may instead lead to a feeling of isolation which would imply more sedentary time spent by the desk. When dichotomizing the group into high and low levels of perceived stress, this association only remained in the high-stress group. These results indicate that social support appears to protect stressed individuals from responding to this stress by adopting unfavorable habits, such as spending too much time sitting. Van der Doef and colleagues also found indications for support being a buffer for unhealthy outcomes [13].

Previous studies have shown that a strained job situation has negative effects on cardiovascular disease, depression and stress-related ill-health [11,12]. Individuals with high-strain and passive jobs have higher odds of being and becoming physically inactive during leisure time [16,17]. Physical inactivity itself, without including job-strain, has many negative effects on health outcomes and mortality [1]. Therefore, a high-strain situation can lead to multiple ill-health related effects. The results of our study indicate that more physical activity and less sedentary time may help to handle job situations with high demand and low support. Furthermore, previous studies have shown that social support can work as a buffer by reducing the unhealthy effect of high strain jobs [13,14].

Limitations to our study include that the analyses are based on a rather small sample with cross-sectional data which means that causality cannot be inferred. We do not know if the physical activity helps individuals to cope with stress-related situations at work, or if it is the stress-related situations that have a spillover effect resulting in less physical activity due to lack of time or reduced capacity for behavioral self-regulation [29]. Another limitation is the inability of the accelerometer to detect all types of physical activity, e.g., standing activities of low intensities or non-ambulatory physical activity. Moreover, length was not measured and therefore BMI could not be calculated or adjusted for in the models. However, one major strength of our study is the use of objective measurement methods for physical activity and sedentary time. More research is needed to assess the relationship between perceived stressors at work and objectively measured physical activity and sedentary time.

## 5. Conclusions

The cross-sectional associations reported in the current study support the hypothesis that more physical activity and less sedentary time may help to handle job situations with high demand and low support. These findings highlight the importance of evaluating how interventions increasing high-intensity physical activity and/or reducing sedentary time may affect job stressors in terms of job demand, control, and support. Moreover, since the causality may be reversed, it is equally important to assess effects of interventions increasing support and decreasing demand on physical activity patterns among office workers.

## Figures and Tables

**Figure 1 ijerph-16-03370-f001:**
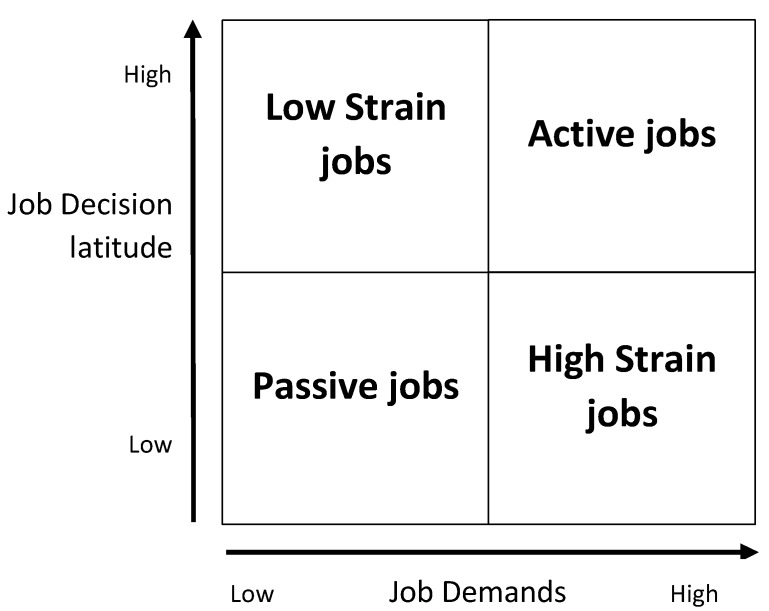
Job demand control model by Robert Karasek [9,10].

**Table 1 ijerph-16-03370-t001:** The distribution of the sample, total *n* = 314.

**% MVPA (*n* = 308)**	
Mean (SD)	6.6 (2.4)
Minimum–maximum	1.7–17.22
**%VPA (*n* = 308)**	
Mean (SD)	1.1 (1.2)
Minimum–maximum	0.0–7.9
**% SED (*n* = 289)**	
Mean (SD)	60.0 (8.3)
Minimum–maximum	34.4–82.4
**Demand, index (*n* = 309)**	
Mean (SD)	3.0 (0.8)
Minimum–maximum	1.0–5.0
**Control, index (*n* = 305)**	
Mean (SD)	3.6 (0.7)
Minimum–maximum	1.3–5.0
**Support, index (*n* = 307)**	
Mean (SD)	4.0 (0.7)
Minimum–maximum	1.7–5.0
**Work engagement (*n* = 314)**	
Mean (SD)	4.0 (0.9)
Minimum–maximum	1.0–5.0
**Stress single item (*n* = 314)**	
Mean (SD)	2.67 (1.2)
Minimum–maximum	1.0–5.0
More rarely or never, *n* (%)	53 (16.9)
Once a month, *n* (%)	103 (32.8)
Once a week, *n* (%)	74 (23.6)
Several times a week, *n* (%)	62 (19.7)
Every day, *n* (%)	22 (7.0)
**Overtime (*n* = 314)**	
Mean (SD)	1.9 (1.1)
Minimum–maximum	1.0–5.0
No, *n* (%)	157 (50.0)
1–3 h/week, *n* (%)	82 (26.1)
3–5 h/week, *n* (%)	39 (12.4)
5–10 h/week, *n* (%)	27 (8.6)
> 10 h/week, *n* (%)	9 (2.9)

MVPA = moderate to vigorous-intensity physical activity, VPA = at least vigorous-intensity physical activity, SED = sedentary time, SD = standard deviation.

**Table 2 ijerph-16-03370-t002:** Results from the linear regression analysis.

Physical Activity	Model 1		Model 2		Model 3		Model 4		Model 5		Model 6	
	*β*	*R^2^*	*β*	*R^2^*	*β*	*R^2^*	*β*	*R^2^*	*β*	*R^2^*	*β*	*R^2^*
**MVPA % log (*n*)**												
Demand (303)	−0.091	0.008	−0.107	0.033	−0.114	0.034	−0.117	0.035	−0.125 *	0.035	−0.116	0.046
Control (299)	−0.111	0.012	−0.130 *	0.040	−0.141 *	0.042	−0.147 *	0.043	−0.144 *	0.043	−0.109	0.050
Support (301)	−0.076	0.006	-0.084	0.029	−0.085	0.029	-0.091	0.029	-0.092	0.030	−0.067	0.039
**VPA% log (*n*)**												
Demand (302)	−0.120 *	0.014	−0.132 *	0.042	−0.148 *	0.047	−0.144 *	0.047	−0.145 *	0.047	−0.137 *	0.057
Control (298)	−0.030	0.001	−0.047	0.032	−0.065	0.037	−0.078	0.042	−0.071	0.045	−0.037	0.051
Support (300)	−0.072	0.005	−0.094	0.035	−0.097	0.037	−0.115	0.042	−0.117	0.045	−0.100	0.048
**SED % (*n*)**												
Demand (284)	−0.050	0.003	−0.034	0.067	−0.044	0.069	−0.031	0.073	−0.058	0.079	−0.053	0.083
Control (282)	0.014	0.000	0.021	0.064	0.015	0.065	0.003	0.069	−0.004	0.072	0.027	0.077
Support (282)	−0.099	0.010	−0.098	0.078	−0.099	0.079	−0.130 *	0.090	−0.130 *	0.092	−0.118	0.095

MVPA = moderate to vigorous-intensity physical activity, VPA = at least vigorous-intensity physical activity, SED = sedentary time. Model 1 = unadjusted, model 2 = adjusted for age, gender, and education, model 3 = adjusted for age, gender, education, and management, model 4 = adjusted for age, gender, education, management, and stress, model 5 = adjusted for age, gender, education, management, stress and overtime, model 6 = adjusted for age, gender, education, management, stress, overtime, and work engagement. * *p* < 0.05. *β* = standardised beta.

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
