# Peer review of "Job Demand-Control-Support Model as Related to Objectively Measured Physical Activity and Sedentary Time in Working Women and Men"

_ijerph, 2019, doi:10.3390/ijerph16183370_

Round 1
Reviewer 1 Report
Please address point 4 in the paper and cite the evidence.
Point 4: BMI must be included as a potential confounder.
Response 4: BMI is not believed to interact with the JD-S-C, and therefore not suitable
as a confounder. Also, the relation between physical activity and BMI is usually very
weak, which limits its effect as a confounder. Therefore, we have not measured length
in our participants and can not calculate BMI.
Author Response
Thank you for reviewing our manuscript (ijerph-598825) and for relevant comments. Please note that the lines we refer to in the responses in the attachment is for the option “Simple Markup” in “Review” in Word.

Reviewer 2 Report
This round of revisions has further improved the manuscript. I only have one remaining comment: The manuscript should present the internal consistency of the scales within the current sample, as opposed to those calculated in previous studies. In case it is needed, here is a useful tutorial I found about how to calculate Cronbach's alpha in SPSS: https://statistics.laerd.com/spss-tutorials/cronbachs-alpha-using-spss-statistics.php
Author Response
Thank you for reviewing our manuscript (ijerph-598825) and for relevant comments. Please note that the lines we refer to in the responses in the attachment is for the option “Simple Markup” in “Review” in Word.

This manuscript is a resubmission of an earlier submission. The following is a list of the peer review reports and author responses from that submission.
Round 1
Reviewer 1 Report
Other cofounders such as BMI, smoking, alcohol, and chronic condition should be included in the analysis.
Please address potential mechanisms
Please expand on limitations including the limitation of accelerometry i.e., cannot account for non-ambulatory PA/Exercise.
Check spelling
i.e.
To be more physical active and spend less time sedentary may help to handle job situations with high demand and low support.
To be more physicallyactive and spend less time sedentary may help to handle job situations with high demand and low support.
Reviewer 2 Report
This study examined the associations between job demand, control, and support and various intensities of physical activity/sedentary behavior in a sample of office-based employees. The authors have a relatively large set of objectively measured physical activity and sedentary behavior, which provides a great deal of potential for making a meaningful contribution to the literature. However, I believe the current analyses and results do not take full advantage of the data available. Some overarching concerns as well as specific comments are outlined below.
General comments:
At the end of the introduction, the specific aims of the study are not well articulated. Which previous findings are inconclusive and what gap is this study aiming to fill? The second part of the purpose statement (“and the mechanisms…”) is quite vague and does not give readers an indication of how these mechanisms will be examined. It could be rephrased to something such as “and the extent to which these associations were moderated by variables such as…” to be more clear. The hypotheses should be updated to match the analyses conducted. For example, job strain situation and active job situation were not assessed or included in the analyses. Similar to the comment above, hypothesis 3 is too vague. The authors should specify how they anticipate these variables will affect the relationships. It would be ideal to identify a few specific hypotheses (e.g., work engagement will moderate the relationship, such that…). As currently written, it comes across as a general exploration for any significant results. More details are needed regarding the control, demand, and support items on the questionnaire, especially since these are the primary variables under investigation. How many items were included for each subscale? How were the items rated? Were the questions specific to control, demand, and support at work? Is there evidence for the reliability and validity of these subscales? If there were multiple items, did the authors check the internal consistency of the subscales? The discussion section makes some conjectures about why certain relationships were observed (or not), and it is important to know how the variables were assessed to determine whether these speculations are plausible. What is the rationale for examining vigorous PA as a separate outcome, especially given the low prevalence of VPA? Alternatively, did the authors consider examining light PA or standing time, which would be more common among these office-based employees? The data analysis seems overly simplistic. When the JD-C-S model was presented in the introduction, the interactions between components (i.e., control relative to demand) were emphasized. Thus, I’m not sure it makes sense to treat them purely as independent predictors in the regression analyses. Furthermore, was multicollinearity a concern? What were the associations between demand, control, and support? More details are needed about the mediation and moderation analyses. The authors clearly state how moderation analyses were conducted with dichotomized variables for stress and support. Similar detail is needed for other variables. For example, how were age, education, overtime work, or work engagement dichotomized (if at all)? What mediation analyses were conducted? Again, these should align with hypotheses – which variables were expected to mediate which relationships? The manuscript would benefit from further English language editing, as there are a number of grammatical issues (e.g,. subject-verb agreement, tense, etc.) throughout.Specific comments:
p. 2, lines 53 and 54: Passive jobs and low-strain jobs were not previously defined. It might be useful to include a 2x2 conceptual model illustrating the possible combinations of low/high strain and low/high control.
p. 2, line 55: Please specify the time frame of the follow-up
p. 2, line 61: Recommend using “non-physically active employees” instead
p. 3, line 98: Please include the specific answer options for the overtime work question
p. 3, line 120: Was sleep time excluded from the totals?
p. 3, line 136: Please clarify how the sample was reduced from n=2024 to n=314. Provide specific reasons why participants were excluded (e.g., X did not meet eligibility criteria, X were not interested in participating, X did not have valid accelerometer data…)
p. 7, line 173: Was “working hard and fast” the way job demand was explained or conceptualized in the measure?
p. 7, line 176: The analysis described from the Hansen study was more complex and had different outcome variables. Please clarify how the results of this study are “in line with” these.
p. 7, lines 186-188: Did the analyses show that stress was associated with sedentary time? The results only report that stress levels moderated the relationship between sedentary time and support.
p. 7, line 193: Similar to the comment above, was the direct association between work engagement and PA examined?
p. 7, lines 197 and 203: Previous studies have shown
p. 8, lines 216 and 218: I’m not sure motivate is the right word here. Consider something such as “suggest it is important to…”
Round 2
Reviewer 1 Report
This statement is not true.
Sitting time has not been associated with stress symptom [5] but lower levels of physical health were observed in a group with more time spent sitting [6].
See:
Teychenne M, Costigan SA, Parker K. The association between sedentary behaviour and risk of anxiety: a systematic review. BMC Public Health. 2015;15(1):1
Table 1 should be a supplement table (S1)
There needs to be a Demographic Characteristics of the Participants table included. Give characteristics of study participants (eg demographic, clinical, social) and information on exposures and potential confounders.
BMI must be included as a potential confounder.
Reviewer 2 Report
The authors have addressed some of my initial concerns. In particular, I appreciate the addition of Figure 1 and Table 1. Overall, the authors' changes were not substantive, though they did provide some justification for not updating the analyses. Many of the explanations and justifications were included in the response to the reviewers' comments, but not in the revised manuscript. I believe it is important to include this information in the manuscript as well, as readers will likely have the same questions. Specific recommendations are outlined below. 1. The hypotheses still need some updating. Hypotheses 1 and 2 are essentially stating the same thing - isn't high demand being negatively associated with PA the same as low demand being positively associated with PA? They are opposite ends of the same spectrum. The authors could instead present separate hypotheses for each variable (demand, control, and support), OR state the hypotheses in terms of interactions between the variables, since these were indeed tested and expected. I would also recommend stating the hypotheses in future tense (e.g., will be associated or would be associated). 2. Hypothesis 3 is still too vague to be considered a hypothesis. Perhaps the authors could instead frame this as an "exploratory aim" of the study if there were no specific moderating or mediating effects anticipated. 3. The authors still do not provide any evidence of reliability or validity of the JD-C-S subscales. At a minimum, they should report the internal consistency (alpha) for each of the subscales, since mean scores were calculated. If the scales have not been previously used or validated, this should be added to the limitations section. 4. The information about how sleep time was excluded from the activPAL totals should be included in the manuscript. 5. In the data analysis section, the authors should clarify what they mean by "all components of the JD-C-S model" and clearly state the interaction terms that were tested in the analyses. 6. The manuscript should clearly explain how the authors selected which mediation and moderation analyses to conduct, and should clearly state which analyses were ultimately conducted. 7. Regarding Point 9 from the first review, if stress was not associated with sedentary time, the last two sentences of paragraph 3 in the discussion should be updated to highlight how social support might buffer the negative effects of high stress on sedentary behavior, rather than simply discussing the direct relationship between stress and sedentary behavior. 8. The added limitation about accelerometers should say "inability" to detect... 9. Regarding Point 12 from the first review, the second "motivated" should also be changed (p. 9, line 231).